# Meta-Analysis of MS-Based Proteomics Studies Indicates Interferon Regulatory Factor 4 and Nucleobindin1 as Potential Prognostic and Drug Resistance Biomarkers in Diffuse Large B Cell Lymphoma

**DOI:** 10.3390/cells12010196

**Published:** 2023-01-03

**Authors:** Mostafa Ejtehadifar, Sara Zahedi, Paula Gameiro, José Cabeçadas, Maria Gomes da Silva, Hans C. Beck, Ana Sofia Carvalho, Rune Matthiesen

**Affiliations:** 1Computational and Experimental Biology Group, iNOVA4Health, NOVA Medical School, Faculdade de Ciências Médicas, NMS|FCM, Universidade Nova de Lisboa, 1169-056 Lisboa, Portugal; 2Departament of Hematology, Instituto Português de Oncologia, 1099-023 Lisbon, Portugal; 3Centre for Clinical Proteomics, Department of Clinical Biochemistry, Odense University Hospital, DK-5000 Odense, Denmark

**Keywords:** annexinA5, diffuse large B cell lymphoma, germinal center, light zone, interferon regulatory factor 4, mass-spectrometry, nucleobindin1, proteomics

## Abstract

**Simple Summary:**

Mass spectrometry-based proteomics studies have suggested various proteins as diagnostic, prognostic, and druggable targets in diffuse large B cell lymphoma. Given rather divergent results in previous mass-spectrometry-based studies, we aimed to review these studies to obtain a consensus list of proteins. From these, we extracted the most consistently significantly regulated proteins. Interferon regulatory factor4, annexinA5, and nucleobindin1 were identified as the most consistently reported proteins. Interferon regulatory factor 4 was identified as a target in immunohistochemistry, proteomics, and genomics research studies. AnnexinA5 and nucleobindin1 appeared as the top up-regulated proteins that may be involved in drug resistance. Additionally, nucleobindin1 and interferon regulatory factor 4 showed a correlation with inferior prognosis based on transcriptomics data. The results of functional enrichment analysis showed that proteins observed as dysregulated in reviewed studies displayed functions occurring in the light zone of the germinal center (e.g., immune synapse formation, and cell migration).

**Abstract:**

The prognosis of diffuse large B cell lymphoma (DLBCL) is inaccurately predicted using clinical features and immunohistochemistry (IHC) algorithms. Nomination of a panel of molecules as the target for therapy and predicting prognosis in DLBCL is challenging because of the divergences in the results of molecular studies. Mass spectrometry (MS)-based proteomics in the clinic represents an analytical tool with the potential to improve DLBCL diagnosis and prognosis. Previous proteomics studies using MS-based proteomics identified a wide range of proteins. To achieve a consensus, we reviewed MS-based proteomics studies and extracted the most consistently significantly dysregulated proteins. These proteins were then further explored by analyzing data from other omics fields. Among all significantly regulated proteins, interferon regulatory factor 4 (IRF4) was identified as a potential target by proteomics, genomics, and IHC. Moreover, annexinA5 (ANXA5) and nucleobindin1 (NUCB1) were two of the most up-regulated proteins identified in MS studies. Functional enrichment analysis identified the light zone reactions of the germinal center (LZ-GC) together with cytoskeleton locomotion functions as enriched based on consistent, significantly dysregulated proteins. In this study, we suggest IRF4 and NUCB1 proteins as potential biomarkers that deserve further investigation in the field of DLBCL sub-classification and prognosis.

## 1. Introduction

Lymphomas are a group of hematologic malignancies that most often grow in lymph nodes [1] and are traditionally divided into two main groups, Hodgkin’s lymphoma (HL) and non-Hodgkin’s lymphoma (NHL). Among all types of NHLs, Diffuse Large B-Cell Lymphoma (DLBCL) is the most frequent subtype characterized by clinical and biological heterogeneity. More than 40% of the DLBCL patients will develop relapsed or refractory disease [2,3]. Gene expression profiling sub-classified DLBCL into three categories: activated B-like (ABC), germinal center B-cell like (GCB), and non-classifiable sub-group [4]. The germinal center (GC) of lymphoid organs is the primary structure where antigen-activated B lymphocytes diversify their immunoglobulin genes through somatic hypermutation (SHM) to produce high-affinity antibodies. The majority of the cells also undergo class-switch recombination (CSR), which produces antibodies with distinct effector characteristics [5]. Continuous activity of downstream molecules in B-cell receptor signaling pathways, like NF-κB and MYD88, is one of the main features of the ABC subtype. ABC tumor cells form during the early stages of plasma cell differentiation in the post germinal center [6,7]. Previous studies suggested that ABC-DLBCLs originate from plasmablasts, although recent studies suggest a memory B cell-like phenotype as the origin [8]. The ABC subtype usually displays a poor prognosis, while the GCB subtype is associated with a better prognosis. The transcriptional profile of the GCB subclass is similar to that of light zone (LZ) GC B cells [9], and the tumor cells harbor various mutations affecting chromatin-modifier genes such as CREBBP and EZH2 (enhancer of zeste homolog 2) [5]. In addition to lymph nodes, the DLBCL tumor cells can be seeded in the testis, breast, uterus, skin, and other organs [10,11]. Primary mediastinal B-cell lymphoma (PMBCL) is a rare and distinct form of DLBCL that involves the mediastinum and has less frequent dissemination to distant lymph nodes [12]. Other subgroups of DLBCL include cutaneous DLBCLs [13], leg type DLBCL [14], T-cell/histiocyte-rich large B-cell lymphomas (THRLBCL) [15], and primary central nervous system (CNS) DLBCL which are rarer than the previous mentioned major groups [16,17].

Evaluation of cell morphology [18] supplemented by immunohistochemistry (IHC) [19] constitutes the main methodology to subtyping DLBCL into GCB and non-GCB in a clinical setting [20,21,22]. However, the results are not always accurate [23,24]. Distinct subgroups of DLBCL have a heterogeneous clinical course and response to treatment, so identification of molecular cues may become a vital part of the treatment decision. Currently, advanced molecular technologies, such as multi-omics, facilitate DLBCL classification [25,26] and DLBCL drug resistance prediction [27]. Nevertheless, discrepant results persist. Finding the sources of the diversity is challenging because of the limitations in access to the samples, the laborious procedure of sample collection (biopsy), the heterogeneity of the disease in different locations, double/triple gene rearrangements [28], and similarities between DLBCL and other types of NHLs originating from the germinal center (such as Follicular and Burkitt Lymphoma). Additionally, routine molecular-based method utilization is still challenging and laborious.

The World Health Organization’s (WHO) lymphoma categorization strategy is based on the results of histopathology assessment, immunophenotyping (including immunohistochemical staining and flow cytometry phenotyping), molecular and cytogenetic results, and clinical symptoms. Currently, IHC algorithms along with patients’ clinical features serve as the first diagnostic strategy for DLBCL and rule out other possible diseases in patients (Appendix A). Moreover, hemato-oncologists additionally group patients according to other criteria, including: (1) lymphoma stage as assessed by imaging methods including computed tomography (CT) scans and positron emission tomography-computed tomography (PET-CT) and (2) treatment responses, such as complete remission (CR), partial remission (PR), stable and progressive disease (PD) (Appendix A). Furthermore, Cheson et al. [29] proposed recommendations for evaluation, staging, and response assessment of patients with Hodgkin’s lymphoma (HL) and non-Hodgkin’s lymphoma (NHL). However, the clinical classification and stratification methods do not reflect the biology of the disease and are too inaccurate to allow targeted therapeutic choices. Meanwhile, different classification approaches, such as IHC and gene expression profiling, for DLBCL patients’ assessment raised the question about the differences in accuracy. Studies that compared IHC algorithms and gene expression profiling revealed a satisfactory level of consistency of both methods for the DLBCL subtyping [30,31,32], but not for drug susceptibility and prognosis prediction [33,34,35]. Of note, a more precise protocol or perhaps a novel method is required to find more predictive biomarkers. Another way to classify DLBCL is through gene mutation profiles and structural chromosomal alterations. The three most frequent groups of genetic mutations and rearrangements in DLBCL are (1) mutations in genes that are necessary for GC reactions (MYC, BCL2, and BCL6), (2) mutations in genes related to DNA, epigenetic factors, and posttranslational modification (PTM) of histones (EZH2, DICER1, CREBBP, and MLL2) [36], and (3) rearrangements involving the MYC, BCL2 and/or BCL6 genes [37]. Overall, gene expression profiling by itself is unable to account for genetic variation. However, next-generation sequencing analyses of the DLBCL genome, whole exome, or transcriptome have uncovered the DLBCL’s genetic make-up and further clarified its pathogenetic processes [38]. Consequently, recent advanced methods are aimed at subclassifying DLBCL tumor cells into subgroups based on the mutations and signaling pathways that drive DLBCL tumor cells. Examples of such classification systems are two recently published landmark studies [39,40]. Additionally, the LymphGen algorithm is an example of a probabilistic tool with therapeutic implications for DLBCL subtype classification based mutations [41].

Heterogeneity between clinical prognostic indexes in DLBCL has been observed in previous studies [42]. It has been suggested that the integration of molecular features of cancer cells and tumor microenvironments with clinical factors increases the chance of identifying high-risk patients [42]. Additionally, recent emerging pieces of evidence highlight that the combination of the international prognostic index (IPI) scores with gene expression-based predictor scores will increase the ability to identify DLBCL patients at high risk of poor outcomes [43]. In addition to IPI, the National Comprehensive Cancer Network-IPI and revised IPI (R-IPI) are two additional prognostic systems that are frequently used in clinics (NCCN-IPI). In the study by Ruppert et al. [42], 2124 DLBCL patients who had R-CHOP therapy between 1998 and 2009 were assessed to determine the most accurate scoring method for predicting overall survival (OS). According to the authors’ findings, NCCN-IPI has the highest level of precision when predicting OS for DLBCL patients receiving R-CHOP therapy. Notably, combining molecular characteristics of cancer cells and the tumor microenvironment with prognostic systems increases the likelihood of identifying high-risk patients [42]. According to a recently published study, the combination of IPI scores with gene expression-based predictor scores will improve the ability to identify DLBCL patients at high risk [43].

Mass spectrometry (MS)-based proteomics is the method of choice to reliably identify and quantify proteins on a large scale. However, divergences in previous MS-based proteomics reports that attempted to introduce new diagnostic, prognostic, and druggable biomarkers highlight the need for consensus insights into the proteome of DLBCL tumor cells. Consequently, in the present meta-analysis study, we address three key questions:(1)In MS-based proteomics studies of DLBCL, which proteins are consistently reported to be regulated?(2)In which biological processes are these consistently regulated proteins more abundant?(3)Are the most consistently regulated proteins related to the prognosis of DLBCL patients?

We identified 15 proteins reported as significantly regulated across 6 articles. These proteins constitute possible target proteins for improvement of, for example, the Hans algorithm and similar algorithms [44]. We expect our meta-analysis to pave the way to building a risk-adapted management strategy for DLBCL that optimizes the diagnosis, treatment, and follow-up.

## 2. Materials and Methods

### 2.1. Protocols and Registration

This meta-analysis is performed according to the PRISMA reporting recommendations [45] and is registered in the International Prospective Register of Systematic Reviews (PROSPERO-CRD42022301846).

### 2.2. Search Strategy and Selection Criteria

A systematic search was conducted in July 2020 and February 2021 on PubMed, PRIDE Archive, Google Scholar, and ScienceDirect. The collected articles were updated in October 2021 before starting the data mining procedure. Published articles in the English language were included, and Ph.D. theses were excluded. The combination of the following keywords: mass spectrometry, diffuse large B cell lymphoma, and proteomics were applied for extracting articles.

### 2.3. Eligibility Criteria

The inclusion criteria for studies included: (1) patients diagnosed with DLBCL for the first time; (2) patients under treatment (first line, second line, and new therapeutic approaches); (3) patients with a relapsed/refractory form of the disease; (4) DLBCL cell lines; (5) intervention with common and novel regimens of chemotherapy; (6) use of MS methods for proteomics analysis; and (7) studies containing a list of significantly regulated proteins. We excluded articles that only contained protein names without providing any measure of regulatory significance, articles on non-human subjects, non-human models [46,47].

### 2.4. Procedure for Selection of Studies 

Searches, titles, and abstracts screening were conducted by two authors (M.E. and S.Z.), and divergences were reviewed by A.S.C., H.C.B., and R.M. Since MS recently underwent a rapid evolution a filter for the year of publications from 2011 to 2021 was applied to retrieve high-quality data sets. This filter provides studies where a large number of proteins are profiled in an unbiased manner for differential regulation. Finally, 27 publications were included in our meta-analysis (Appendix A).

### 2.5. Data Extraction Process

Two investigators (M.E. and S.Z.) extracted the data based on the following strategy:All protein lists from the 27 articles were extracted, including the identification codes, level of regulations, and *p*- or adjusted *p*-values.Significantly regulated proteins (*p*- or adjusted *p*-value < 0.05) were separated. We chose this simplified strategy because all the studies explored the same broad question, i.e., “how is the proteome of DLBCL tumor cells changed by chemotherapy?” or “what are the differences between the proteomes of DLBCL subtypes?”. Because of the wide variety of conditions used in the selected articles, our method does not involve pooling the numerical estimates of the conditions’ effects and sensitivity analysis.Features of the Uniprot website (https://www.uniprot.org/; accessed on 1 October 2021) were used to unify protein identification codes.Significantly regulated proteins extracted from 27 papers were compared to identify consistent proteins.For the most consistent proteins, the regulation of proteins was converted from ratio/normalized ratio/Z-value/log_2_ ratio/log intensity to the fold change. Then proteins with log_2_ fold change <0 were considered as the down-regulated (−1) and 0 < log_2_ fold change as the up-regulated proteins (+1) if the indicated *p*-value were significant.

### 2.6. Assessment of the Risk of Bias in Individual Studies

The risk of bias for each study was assessed by two independent reviewers (M.E. and S.Z.). The assessed criteria included: (1) bias arising from the randomization process; (2) bias due to deviations from the intended intervention; (3) bias due to missing outcome data; (4) bias in the measurement of the outcome; and (5) bias in the selection of the reported results (Appendix A). The results of the risk of bias assessment revealed one study with a high risk [48], seven with a low risk [27,49,50,51,52,53,54], and nineteen with a moderate risk of bias [26,55,56,57,58,59,60,61,62,63,64,65,66,67,68,69,70,71,72].

### 2.7. Data Management and Statistical Analyses

The final list of the consistent proteins was divided into three lists of proteins that were identified by four to six articles. R studio (version 1.3.1073) was used for data analysis, functional enrichment, and data visualization. The clusterProfiler R package (http://bioconductor.org/packages/release/bioc/html/cluster Profiler.html; version: 4.6.0, Guangchuang Yu, Hong Kong, accessed on 1 January 2022) was used for functional enrichment analysis. In all cases, the proteins from proteomics were based on significantly regulated proteins from 27 studies (Appendix A). All significantly regulated proteins in 27 studies were compared to IHC markers, and the number of articles concurring was estimated. We applied the thresholds below for the different analyses performed:(1)We used significantly regulated proteins identified at least in four articles for comparison of proteomics and genomics and for functional enrichment analysis. In addition, we compared proteins implicated in persistent, unresolved inflammation with proteins that were significantly regulated based on at least four proteomics articles.(2)For the comparison of direction of regulation, evaluation of prognosis and survival analysis, we considered only the most consistently regulated proteins, which were 15 proteins that were identified as significantly regulated in 6 articles out of the 27 studies.

## 3. Results

The reviewed articles applied different types of MS techniques and sample types. In terms of the methodology, they mainly used proteomics technologies such as matrix-assisted laser desorption/ionization MS imaging (MALDI-MSI), high-resolution shotgun proteomics, liquid chromatography MS (LC-MS/MS), gas chromatography MS (GC-MS), LC/GC–tandem MS, matrix-assisted laser desorption ionization-time of flight MS (MALDI-TOF MS), etc. The sample types that have been investigated in the reviewed articles include formalin-fixed paraffin-embedded tissue (FFPE), cell lines, cerebrospinal fluid (CSF), serum, and frozen tissues (Appendix A).

### 3.1. General Overview of Extracted Proteins 

We collected regulated proteins from 27 proteomics and MS-based studies addressing poor prognosis in DLBCL (Appendix A). From collected proteins, 206, 45, and 15 were identified as significantly regulated in four, five, and six articles, respectively (Figure 1, Appendix A). The 15 most consistently identified proteins from 6 articles were ALDH18A1, CDK6, IRF4, SOD1, B2M, ARPC5, PRMT1, CD44, CPSF7, EZR, COPG1, IGHG1, SCRN1, ANXA5, and NUCB1 (Appendix A).

### 3.2. Concordance between IHC and Genomics with the Most Consistently Identified Proteins

The appearance of IRF4 among the most consistently regulated proteins encouraged the search for other markers of the Hans algorithm and similar algorithms (CD10, BCL6, FOXP1, GCET1, SERPINA9, and LMO2) [73]. LMO2 (P25791) and GCET1 (SERPINA9- Q86WD7) proteins were not among the reported regulated proteins in MS studies. FOXP1, MME (CD10- P08473), BCL6, and IRF4 were identified in 2, 3, 3, and 6 papers, respectively (Figure 2A). IRF4 was identified as significantly regulated by proteomics MS-based studies more frequently than other Hans algorithm markers. In normal GC, the high level of BCL6 in DZ reduces the cell susceptibility to DNA damage, which is necessary for the high proliferation rate, clonal expansion, and somatic hypermutation. Also, BCL6 represses IRF4 expression in DZ to block B-cell differentiation. In B-cells while exiting LZ, proteins like IRF4 and later PRMD1 suppress the expression of BCL6. Additionally, FBXO11 degrades the remaining BCL6 [5,74,75].

Moreover, the biological source of the samples and sample preparation methods for MS are possible reasons for the absence of Hans markers in MS results. MS-based studies analyzed cell lines, CSF, serum, FFPE tissue, and frozen tissue. FFPE tissue is the most conventional type of sample for IHC assessment. 

Based on comparative data from FFPE and FF tissue, the practicality and, in some cases, superiority of FFPE over fresh frozen (FF) tissue has been reported for mutation assessment, protein extraction, and mRNA sequencing [66,76,77]. Four out of the six studies identified IRF4 in cell lines. In addition, two studies identified IRF4 in frozen tissue and FFPE samples.

Subsequently, proteins from proteomics that were reported as significantly regulated by at least four articles (Appendix A) and genomics-based studies [78] were compared. Both omics methods reported several common proteins, including IGHM, IRF4, NF-kB, MEF2B, PAX5, VCL, SAMSN1, BST2, IL4I1, PTPN1, REL, IL-16, and CARD11 (Figure 2B). ITPKB, SLAM, and TNFAIP3 were mutated in multiple sub-types but were not regulated in proteomics studies (Figure 2B). A subset of these proteins are GC-dependent proteins such as IRF4, NF-kB, MEF2B, PAX5, IL-16, REL, and CARD11 that are reported in 6, 5, 5, 4, 4, 4 and 4 articles, respectively. These proteins are necessary for GC formation, GC maintenance, and the late differentiation of B-cells. LymphGen is a recently developed classification of DLBCL tumors into six main subtypes (MCD, BN2, EZB, ST2, A53, and N1) based on similarities in genetic mutation. Several genes were identified as common between LymphGen subclasses and significantly regulated proteins by proteomics studies (Appendix A). For instance, IRF4, HLA-C, IL-16, and TBL1XR1 from the MCD subtype, MEF2B, GNAI2, and REL from the EZB subtype, B2M and CNPY3 from the A53 subtype, and ALDH18A1 from the N1 subtype (Figure 2C). In conclusion, an overlap between LymphGen target genes and proteins identified in MS-based proteomics studies were observed.

NF-kB, REL, and IRF4 are representative biomarkers for LZ, from where ABC subtype originates [5]. IL-16 emerged as another protein identified in both proteomics as significantly regulated and genomics studies as mutated (Figure 2B). IL-16 is a chemotactic factor derived from B-cells. IL-16 attracts CD4^+^ T helper lymphocytes and CD4^+^ dendritic cells (DCs) toward the B-cell-rich follicles [79].

### 3.3. Regulation of the Most Consistently Reported Dysregulated Proteins

The following steps were applied to define the up-or down-regulation of proteins: (1) protein regulations were standardized to log_2_ fold change; (2) the direction of all regulations was converted to use the control groups (defined as the group most similar to favorable prognosis) as a reference for the comparisons; (3) log_2_ fold change was regarded as up-regulated if above zero and with a significant *p*-value, which was assigned a score of 1 and (4) log_2_ fold changes below zero and significant were considered downregulated and assigned a score of −1. Non-significant regulated proteins were not considered. Finally, the scores for each protein were summed to obtain a measure of overall consensus across studies. The results indicated overall up-regulation of SCRN1, ARPC5, PRMT1, COPG1, IGHG1, B2M, ANXA5, and NUCB1 and downregulation of EZR, CPSF7, ALDH18A1, CDK6, IRF4, SOD1, and CD44 when comparing the poor prognosis group to the group with favorable prognosis (Appendix A). From the 15 most consistently reported dysregulated proteins, mutations in B2M, ALDH18A1, and IRF4 were reported in the A53, N1, and MCD subtypes of DLBCL, respectively. ALDH18 is the most consistently down-regulated protein in proteomics studies and was previously reported with missense mutations [41]. This may explain the downregulation of ALDH18 in proteomics studies.

In addition, the protein scores revealed that NUCB1 and ANXA5 are the two most consistently up-regulated proteins with the highest scores, and are furthermore correlated with tumors’ drug resistance and invasion. Some of the studies that reported NUCB1 and ANXA5 as up-regulated analyzed the effects of anti-lymphoma drug combinations, such as R-CHOP/CHOP regimens and immunomodulatory agents, on chemo-sensitive or -resistant DLBCL patients’ samples and DLBCL cell lines (Figure 3). We reviewed the literature to obtain deeper insights into the roles of NUCB1 and ANXA5 in DLBCL resistance.

#### 3.3.1. Annexin A5 (ANXA5)

Annexin A5 (ANXA5) is a member of the annexin family that binds to phospholipids, like phosphatidylserine (PS) molecules, in the presence of Ca^2+^. It has been reported that inhibition of ANXA5-PSs binding leads to apoptotic cell ignorance by macrophages. Accumulation of those apoptotic cells will subsequently induce DC immune responses [80,81]. In the tumor microenvironment (TME), administration of ANXA5 can stop the immunosuppressive properties of TME, such as a high level of PD-L1 expression after chemotherapy. On a molecular scale, ANXA5 increases tumor cells’ immunogenicity in different ways. Fusion of ANXA5 with antigenic peptides of tumor cells not only concentrates tumor antigen peptides in TME but also exaggerates the effects of other immune checkpoint inhibitors [82,83,84]. This strengthened the roles of ANXA5 in tumors’ behaviors and responses to chemotherapy. In DLBCL, a comparison between differentially expressed genes (DEGs) of tumor cells and non-cancer samples, which were extracted from Cancer Genome Atlas (TCGA) and the Gene Expression Omnibus (GEO) in the drug-gene interaction database, revealed that ANXA5 is a target for Daunorubicin, a drug similar to Doxorubicin [85]. Furthermore, a pilot study tested the ANXA5 impact on DLBCL cell lines’ invasiveness and resistance to CHOP. Intriguingly, the findings showed that the ANXA5 blockade agent combined with CHOP significantly decreased cell invasion, MMP-9 expression, and chemoresistance to CHOP in a PI3K-dependent mechanism [86]. Currently, the role of ANXA5 is controversial. On the one hand, it has been established that ANXA5 increases tumor cell immunogenicity when fused with tumor antigens, whereas in DLBCL *in vitro* models, ANXA5 inhibition sensitizes tumor cells to CHOP.

#### 3.3.2. Nucleobindin1 (NUCB1)

Nucleobindin1 (NUCB1) is a member of the nucleobindin family, and the nucleobindin family is classified into the calcium-binding proteins (CBPs) group. Much is known about other CBPs such as calmodulin and calretinin, but few reports have focused on the biological functions of the NUCB family. NUCB1 presence has been detected in the supernatant of a murine B-cell line with systemic lupus erythematosus. Further investigations identified NUCB1 molecules around the Golgi apparatus. In this location, NUCB1 proteins are one of the major Ca2^+^-binding proteins [87,88,89,90,91,92]. Also, NUCB1 is a part of the endoplasmic reticulum (ER) responses to natural and chemical stressors. Under stress conditions, NUCB1 promoter activity will be increased, resulting in a high amount of NUCB1 inside and outside of the cell. The increased level of NUCB1, especially near the Golgi apparatus, inhibits activating transcription factor 6 (ATF6) via blocking its interaction with SP1 and results in unfolded protein response (UPR) suppression; therefore, induced apoptosis through UPR can be hampered by an increased level of NUCB1 [93]. In genome-based studies that analyzed genetic features of the DLBCL, NUCB1 was not detected. However, proteins related to endoplasmic reticulum stress are mutated. For instance, XBP1, PRC2C, DDX3X, and SGK1. In the LymphGen algorithm, XBP1, PRC2C, DDX3X, and SGK1 were identified in the ST2 subtype of the DLBCL, which suggests an association with endoplasmic reticulum stress and the ST2 subtype [41,94].

*In vivo* experiments revealed that NUCB1 was cleaved by programmed cell death induction [95] in the cell cytoplasm and caspase −3, −6, and −8 mediate the cleavage of NUCB1 and NUCB2 [95]. However, the effects of other caspases cannot be completely ruled out. In addition to the cytoplasm and extracellular space, the presence of NUCB1 in the nucleus has raised questions about its role in gene expression. The interactions between NUCB1 and the E-box motifs of certain genes involved in tumorigenesis strengthen the hypothesis of NUCB1’s role in cancer progression [96]. NUCB1 mRNA expression has previously been found to be increased in transformed high-grade NHL [97]. In line with these reports, NUCB1 was used as a cancer marker for colon cancer [98]. Although it has been demonstrated that NUCB1 overexpression helps tumor progression [96], a 5-year follow-up study showed the worst prognosis for pancreatic ductal adenocarcinoma (PDAC) patients with low levels of NUCB1. *In vivo* and *in vitro* validation confirmed that overexpressed NUCB1 protein inhibits cell proliferation and by ATF6 regulation, NUCB1 increases drug sensitivity of cancer cells [99]. Normal and cancerous tissue extracellular matrix (ECM) turnover is another process to which NUCB1 contributes. NUCB1 is a binding partner for matrix metalloproteinases (MMPs), particularly MMP2 and MT1-MMP. Alive cell visualization revealed a NUCB1-dependent secretory pathway for MMPs from the ER to the Golgi, and then extracellular space. It seems that invasive migration of normal and cancerous cells and ECM degradation of NUCB1 play a role indirectly by controlling MMP secretion levels [100]. Although NUCB1’s relevance in DLBCL has not previously been addressed, previous findings on other tumor cells revealed no consensus about NUCB1’s functions in tumor progression.

### 3.4. Correlation of NUCB1, ANXA5, and IRF4 mRNA Expression to DLBCL Patients’ Survival

The mRNA expression from Affymetrix chips in different stages of B-cell differentiation of the 15 most consistently identified proteins was extracted from the Genomicscape (http://www.genomicscape.com; accessed on 1 July 2022) website in the GS-DT-58 dataset. Based on the results, CPSF7 and IGHG1 showed the lowest and highest levels of mRNA expression levels during normal B-cell differentiation, respectively. The results of the mRNA expression analysis showed that the most consistently regulated proteins observed in our meta-analysis were all expressed in normal B-cells, from naïve stage to fully differentiated bone marrow–established plasma cells (Figure 4A–C and Appendix A). While NUCB1 was the second lowest in terms of mRNA expression level in normal B-cells, in all 9 conditions of 6 MS-based studies it appeared to be a significantly up-regulated protein. In normal B-cells, IRF4 was the third most expressed mRNA after IGHG1 and B2M. Also, the mRNA expression levels of IRF4, NUCB1, and ANXA5 in different subtypes of the DLBCL based on the publicly available data on Genomicscape were analysed (dataset: GS-DT-20). The results revealed that IRF4 and NUCB1 mRNA levels are higher in the ABC subtype than in the GCB subtype. On the other hand, ANXA5 mRNA is more expressed in GCB than in the ABC subtype. Notably, the differences between tumor and normal cells in NUCB1 mRNA expression were more pronounced than for IRF4 and ANXA5 (Figure 4D–F). Extracted data here showed the importance of the IRF4, NUCB1, and ANXA5 proteins as potential diagnostic and classification biomarkers. Next, the patient survival based on mRNA expression levels of the 15 most consistently identified proteins in the Genomicscape (dataset: GS-DT-20) was evaluated. Among all transcripts, ANXA5 and CPSF3 showed a non-significant *p*-value in the survival analysis. Additionally, a high expression of NUCB1, COPG1, CD44, ALDH18A1, IRF4, SCRN1, B2M, and PRMT1 mRNAs was associated with the worst overall survival (OS) for patients with DLBCL (Figure 4G–I and Appendix A). The results of NUCB1, ANXA5, and IRF4 (by MS, IHC, and genomics) were further explored in GEPIA (http://gepia.cancer-pku.cn) [101]. The analysis of the effects of NUCB1, ANXA5, and IRF4 mRNA expression on the OS of DLBCL patients in GEPIA showed the same prognostic values for the corresponding proteins as the results given by Genomicscape (dataset: GS-DT-20) (Figure 4J–L). An oligonucleotide microarray analysis-based study found that NUCB1 was up-regulated during the transformation from a low-grade NHL (FL) to a high-grade NHL (DLBCL) [97] with an inferior 5-year relative survival rate [102]; therefore, the elevation of NUCB1 can indicate poor prognosis in this setting. Similar to NHL, in colorectal cancer, high expression levels of NUCB1 showed an association with shorter PFS and OS [103]. IRF4 as a terminal transcription factor of B-cells showed the highest hazard ratio (HR) for poor outcomes in comparison with NUCB1 and ANXA5. Previous research [104,105] identified IRF4 as a poor prognostic biomarker in DLBCL and multiple myeloma. However, two studies observed no prognostic value for IRF4 in DLBCL [20,21]. Discrepancies in IRF4 prognostic values might be the result of differences in cutoff values that the authors defined to consider tumor cells positive or negative for IRF4. The prognostic value of ANXA5 is still ambiguous for DLBCL, although its downregulation appears as a poor prognostic marker with a high HR in bladder cancer [106]. Therefore, the usefulness of these three proteins as clinical diagnostic and prognostic markers for DLBCL needs to be clarified.

### 3.5. Overall Functional Analysis of Consistently Identified Proteins

KEGG pathway analysis and GO enrichment analysis were performed to elucidate the functional roles of the consistently regulated proteins in DLBCL. The proteins that were consistently reported as regulated by at least 4 articles (266 gene names generated 265 Entrez ID) were considered in the analysis. GO enrichment analysis by clusterProfiler for biological process (BP), cellular component (CC), and molecular function (MF) of 265 common proteins in at least four articles illustrated neutrophil degranulation, cell-substrate junction, and cell adhesion as the most significantly enriched terms, respectively (Figure 5A–C). In agreement with the previous results, all the enriched terms (cell-substrate junction, degranulation, and cell adhesion) indicate that 265 common proteins are mostly involved in the cell cytoskeleton rearrangement, locomotion, and secretory functions. Furthermore, genome-based approaches demonstrated missense and truncating mutations in cytoskeleton proteins like ACTG1 and ACTB in ST2 subtype [41]. These functions are essential for B-cells in the LZ since they must migrate between the LZ and the DZ and interact with other immune cells to increase their affinity for antigens. The reviewed studies analyzed tumor cells that were exposed to chemotherapy drugs *in vivo* or *in vitro*. Therefore, we conclude that GO enriched terms suggest that DLBCL tumor cells are active in migration and secretion, which requires further follow-up studies. KEGG pathway analysis revealed that EBV infection-associated proteins are the most significant pathway with the highest number of proteins (Figure 5D and Appendix A). In the EBV infection pathway immune related proteins with antitumor and tumorigenic function were identified, e.g., NF-kB, MHC-I, PI3K, B2M, Bax, CD40, and CD44 (Appendix A). These proteins could potentially contribute to unresolved inflammations as a result of viral infection [107]. Although, EBV infection is only present in a minor fraction of patients. ClusterProfiler also identified p53, calcium, and NF-κB signaling pathways as enriched. In addition, KEGG analysis identified enrichment in Fc gamma R-mediated phagocytosis, antigen processing and presentation, and leukocyte transendothelial migration, which are features of GC B-cells, particularly in LZ. B-cells in GC present antigens to the T follicular helper (Tfh) cells to survive and pass the selections [9]. Therefore, the result of the KEGG functional analysis identified functions related to B-cell activities that form immune synapses with other immune cells, which is an important feature of GC-LZ.

## 4. Discussion

DLBCL patients usually experience long-term disease-free survival ranging from 60–70%. However, 30–40% of DLBCL patients suffer from a recurrence of the disease or are refractory to treatment. Unfortunately, the prognosis for refractory/resistance (R/R) patients is poor and 80% of these patients will die, even after modern salvage therapies, autologous stem cell transplantation [108], and even CAR T cell treatment. This highlights the importance of improvements in currently used methods for DLBCL diagnosis and stratification to improve the prediction of refractoriness, recurrence, and death, if possible. Promising findings from recent proteomics and MS-based studies have opened a new way to analyze DLBCL tumor cells that may solve inconsistencies among IHC algorithms and gene expression data. The majority of the 27 studies reviewed here used LC-MS/MS and an Orbitrap mass analyzer. Proteomic technologies applied in cancer biomarker discovery, mainly based on liquid chromatography coupled to tandem mass spectrometry, is being employed more extensively [109,110].

We reviewed the literature of MS-based studies aimed at identifying consistent prognostic and druggable biomarkers in DLBCL. The reported studies identified protein regulation in various contexts, explaining the diversity of identified proteins (Appendix A). Nevertheless, 15 proteins were consistently identified in at least six papers. We consider that the main variations in markers identified are caused by the different types of samples analyzed (e.g., cell lines, paraffin-embedded tissue, frozen tissue, CSF, and serum). However, it is well established that MALDI and ESI ionization result in coverage of different sets of proteins. It is possible to observe that two instruments with ESI ionization ion source but different mass analyses identify different sets of peptides. However, this does not imply that quantitative data from MS is inaccurate when samples are compared within a study. Nonetheless, the covered proteins found to be regulated vary depending on the experimental design, sample type, instrument, and setting. The comparison of proteomics and genomics revealed several similarities between MS-based proteomics and genome-based studies, although a large-scale study is still needed to compare proteomics and genomics on the same set of DLBCL samples.

IRF4 was the only protein that we found in common between proteomics, IHC, and genomics-based studies [78]. In lymphocytes, IRF4 is one of the rapidly induced downstream genes of Rel/NF-κB and CD40 pathway proteins. IRF4 levels rise in response to stimuli or when IRF4 upstream proteins are overexpressed in cancer [111,112]. In conventional IHC panels, being positive for IRF4 in more than 30% of the cells classifies samples as a non-GCB subtype. According to the LymphGen algorithm, high expression of IRF4 and low expression of Blimp-1 (PRDM1) is a feature of the MCD subtype of the DLBCL [41]. In Deeb et al.’s study on cell lines, they reported a higher level of IRF4 in ABC cell lines in comparison with the GCB cell lines [26]. Furthermore, we previously identified a higher level of IRF4 in ABC versus GCB subtypes, which is in agreement with Deeb. S.J. et al.’s study and IHC algorithms [72]. Besides the IRF4 expression level in DLBCL subtypes, Hagner P.R. et al. showed that IRF4 could be downregulated in DLBCL tumor cells treated with different drugs [50]. This raises the question as to whether treatments might have an impact on IRF4 expression (or other IHC indicators), and if so, if this would have an influence on the efficiency of the IHC categorization.

ANXA5 and NUCB1 were regulated in the nine conditions of six studies. We found that ANXA5 showed up-regulation in three chemosensitivity-related articles. According to Fornecker et al. [27], DLBCL patients that relapsed or were refractory to rituximab plus an anthracycline-based chemotherapy regimen showed significantly higher levels of ANXA5 than those who achieved a complete response with the first line of the treatment. Liu, Y et al. [61] compared patients with low and high sensitivity to CHOP treatment, and they showed ANXA5 up-regulation in frozen tissue samples of patients with low sensitivity to the regimen. Hagner et al. [50] evaluated the effects of CC-122 on lymphoma cells by using MS, and observed slight ANXA5 up-regulation in lymphoma cell lines in the presence of lenalidomide and CC-122. The level of ANXA5 appears to be higher in treated tumor cells than in untreated cells, but the level of ANXA5 in resistant DLBCL tumor cells requires further investigation. Moreover, Deeb et al. [26] showed that ANXA5 expression is up-regulated in ABC versus GCB subtypes.

NUCB1 showed correlations with secretory activities in lymphocytes. Malignant and reactive lymphocytes in NHL and infectious diseases contain more NUCB1 than resting lymphocytes [113]. Our recently released publication, in agreement with Sally J. Deeb et al., confirmed the up-regulation of NUCB1 in ABC subtypes of DLBCL cell lines in comparison with GCB subtypes at the cellular and extracellular levels [26,72]. Clinically, patients suffering from the ABC subtype of DLBCL have an inferior prognosis, and the chance of relapse is higher than in the GCB subtype. Here, the possible molecular role of NUCB1 is unclear. To investigate the biological roles and NUCB1’s importance in drug response, Yong-Qiang Hua et al. [99] carried out a study on a pancreatic cancer cell line and animal models. They demonstrated that NUCB1 overexpression suppresses proliferation and increases susceptibility to gemcitabine. Moreover, data mining in the UALCAN database (http://ualcan.path.uab.edu/analysis.html; accessed on 1 July 2022) revealed that downregulation of NUCB1 correlates with a worse prognosis for patients with pancreatic ductal adenocarcinoma (PDAC) [99], while our analysis showed the opposite effects in DLBCL. Also, Yong-Qiang Hua [99] demonstrated the role of N^6^-methyladenosine (m^6^A) in NUCB1 mRNA modification. Several papers have reported the contribution of m^6^A in regulating different genes in DLBCL, but it is unclear how this modification influences NUCB1 levels in DLBCL [114,115]. Post-transcriptional or -translational modifications might regulate levels of NUCB1 in DLBCL. We found several significant pathways and terms related to protein production and hemostasis, including proteasome in KEGG analysis and translation regulatory, translation repressor activity and mRNA regulatory element binding in GO-MF (Figure 5C,D). Moreover, in survival analysis, NUCB1 and IRF4 showed significant *p*-values and HR > 2, while ANXA5 was not statistically significant with HR = 1.3 for DLBCL survival (Figure 4). The contribution of NUCB1 and ANXA5 in tumor progression and drug responses in DLBCL remains controversial, so revealing their exact roles in DLBCL behavior and prognosis needs further investigation. Furthermore, functional assays exploring the differential expression of NUCB1 and ANXA5 in ABC and GCB subtypes remain for future studies.

A limitation of our meta-analysis is the current lack of proteomics data from the recent sub-classifications of GCB and ABC proposed by Chapuy et al. [39] and Wright et al. [41]. These studies demonstrated that DLBCLs can be subclassified into more than five molecularly defined groups. Given that these novel classifications are widely recognized and acknowledged both in the research field and in clinical practice, future research targeting the proteome of these molecular entities may provide novel biomarkers and drug targets.

The functional enrichment analysis revealed that tumor cells, even under chemotherapy treatment, contain proteins involved in processes like migration, cytoskeleton locomotion, and cell adhesion (ICAM, plexin, IL-16, CD40, ARPs, and ezrin). For B-cells in LZ, cytoskeleton reformation and secretion activities are two necessities, since B-cells are required to migrate between DZ and LZ and communicate with resident cells. In this line, the presence of CD40, CD86, and CD83 and the lack of CXCR4 [9] among the significantly regulated proteins identified by MS-based studies reflects partly the proteins in LZ. In LZ, lymphocytes come into contact with other immune cells like DCs and Tfh to form immune synapses. In immune synapses, B-cells and other immune cells affect each other with insoluble and soluble molecules [116]. Interestingly, the appearance of IRF4 among the 15 most consistently regulated proteins strengthens this idea. Mainly, because IRF4 is constitutively expressed in the selection and class-switch recombination steps in LZ [7].

In Appendix A, we identified common proteins among significantly regulated proteins based on at least four proteomics articles and proteins related to imbalance and prolonged inflammation, for example interleukins, SOD enzymes, and kinases [117]. A majority of these proteins are immune response-related proteins. Inflammation triggers crosstalk between two highly conserved biological processes: apoptosis and wound healing, which keep immune responses under control [117]. Toll-like receptors, interleukins, TNF, histamine, SOD enzymes, growth factors, catalase, and kinases are examples of immunomodulatory proteins involved in immune response control. Persistent, unresolved inflammation disrupts the balance of apoptosis and wound healing, increasing the risk of pathological conditions such as cancer [117,118,119,120]. DLBCL, like other cancers, could be the result of an imbalance in immune control. Previous research has found that genetic background (such as HLA zygosity status) [121], autoimmune diseases, and viral infections all increase the risk of DLBCL [107,122]. Furthermore, genetic factors influence the distribution and behavior of immune cells in the tumor microenvironment [123]. Previous research has shown that immune cell composition plays an important role in the outcome of DLBCL [124,125,126]. However, the crosstalk between the different immune cells is more important than the number of immune cells in the DLBCL tumor cells’ niche. Also, the correlation between immune cells can affect the response to the treatment [127].

Data collection and unification of the results in various papers suggest IRF4, NUCB1, and to a lesser extent, ANXA5 proteins as potential biomarkers that need further investigation in regard to DLBCL prognosis, chemoresistance, and sub-classification. Adding NUCB1 and ANXA5 proteins to the current diagnostic algorithms might improve DLBCL management. No correlation was observed between survival and ANAX5 mRNA expression. However, investigation of ANAX5 protein expression and survival may provide a different result compared to what transcriptomics studies demonstrated. Overall, proteomics-MS-based studies in combination with orthogonal methods may improve our understanding of the DLBCL classification and drug resistance.

## 5. Conclusions

IRF4 and other GC-dependent transcription factors, along with functional enrichment analysis, showed that proteins involved in immune synapse interactions deserve further research in DLBCL. Additionally, we found NUCB1 and ANXA5 as the most consistently identified proteins from 27 articles that were up-regulated in drug resistance DLBCL samples. These proteins should be further assessed by future studies as prognostic and chemoresistance biomarkers. Furthermore, high mRNA expression levels of IRF4 and NUCB1 are poor prognostic markers for patients suffering from DLBCL. Consistently reported MS-based potential markers were observed to partly concur with drug resistance markers obtained by other omics technologies. Large-scale studies with a large sample size would be of particular interest to exploit the full potential of MS and to perceive behind-the-scenes DLBCL drivers.

## Figures and Tables

**Figure 1 cells-12-00196-f001:**
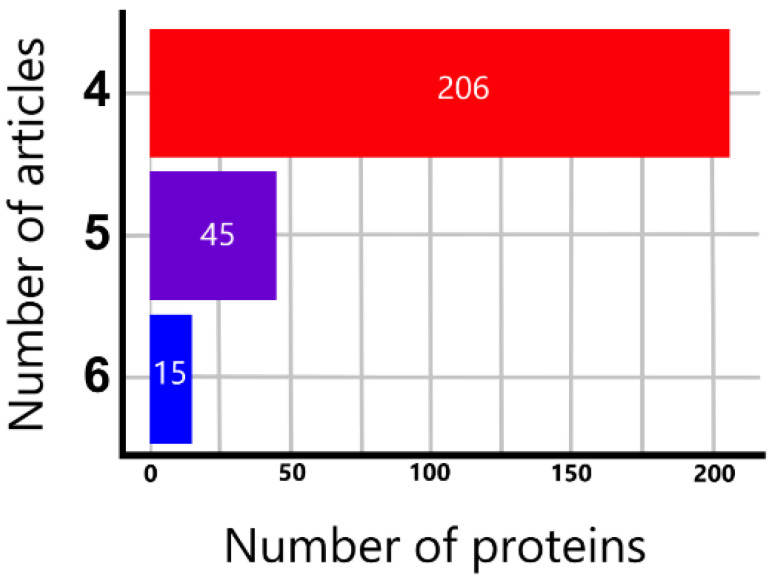
The number of significantly regulated proteins was reported consistently by 4, 5, and 6 MS-based articles.

**Figure 2 cells-12-00196-f002:**
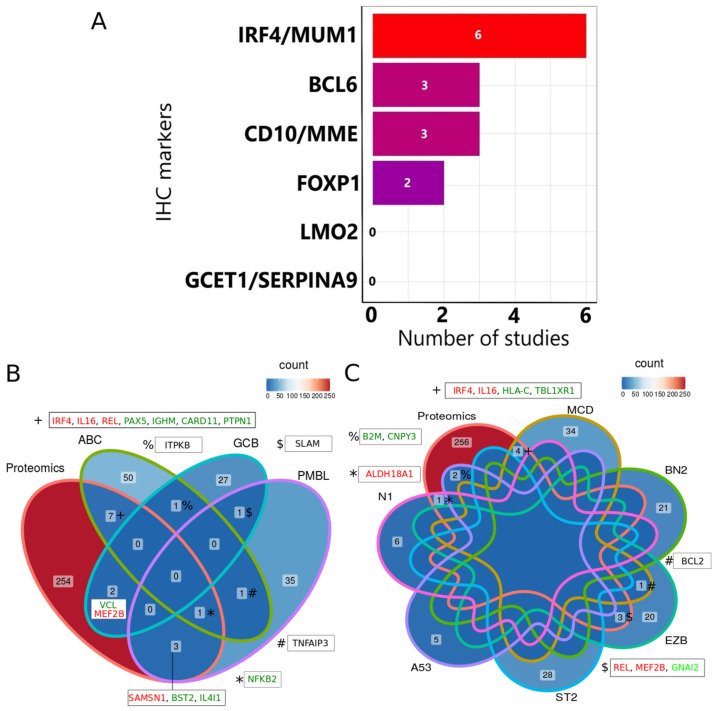
An overview of common proteins/genes between IHC, genomics and proteomics. (**A**) The number of times IHC markers were reported in the 27 reviewed articles. (**B**) Common gene overlap between the current study and genome-wide studies. (**C**) Common genes between the LymphGen algorithm and proteomics. The colors in the Venn diagrams indicate the number of genes in each area. Red and blue represent the area with the highest and lowest number of genes, respectively. In (**B**,**C**) up- and down-regulated proteins common between proteomics studies, genome-based studies, and the LymphGen algorithm are indicated with green and red letters, respectively. Genes mutated in multiple subtypes but not identified as regulated in proteomics studies are indicated with black letters. The symbols “* ^$ * # % +^” links Venn sub group with the indicated gene names.

**Figure 3 cells-12-00196-f003:**
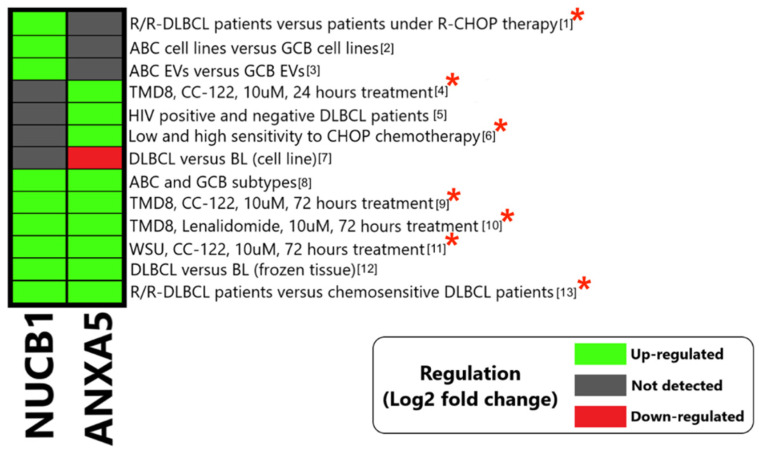
Heatmap summarizing the regulation of the two most consistently identified and up-regulated proteins (ANXA5 and NUCB1) in articles that identified them as the significantly regulated proteins. Colors are coded based on the protein direction of regulation and the score of 0 (black) indicates those studies that did not identify ANXA5 or NUCB1 proteins. The red stars show studies that tested drug effects on samples of DLBCL patients or cell lines. R/R: relapse/refractory, TMD8: Tokyo medical and dental university 8 (cell line), CC-122: cereblon E3 ligase modulator avadomide; WSU: Wayne State University (Appendix A).

**Figure 4 cells-12-00196-f004:**
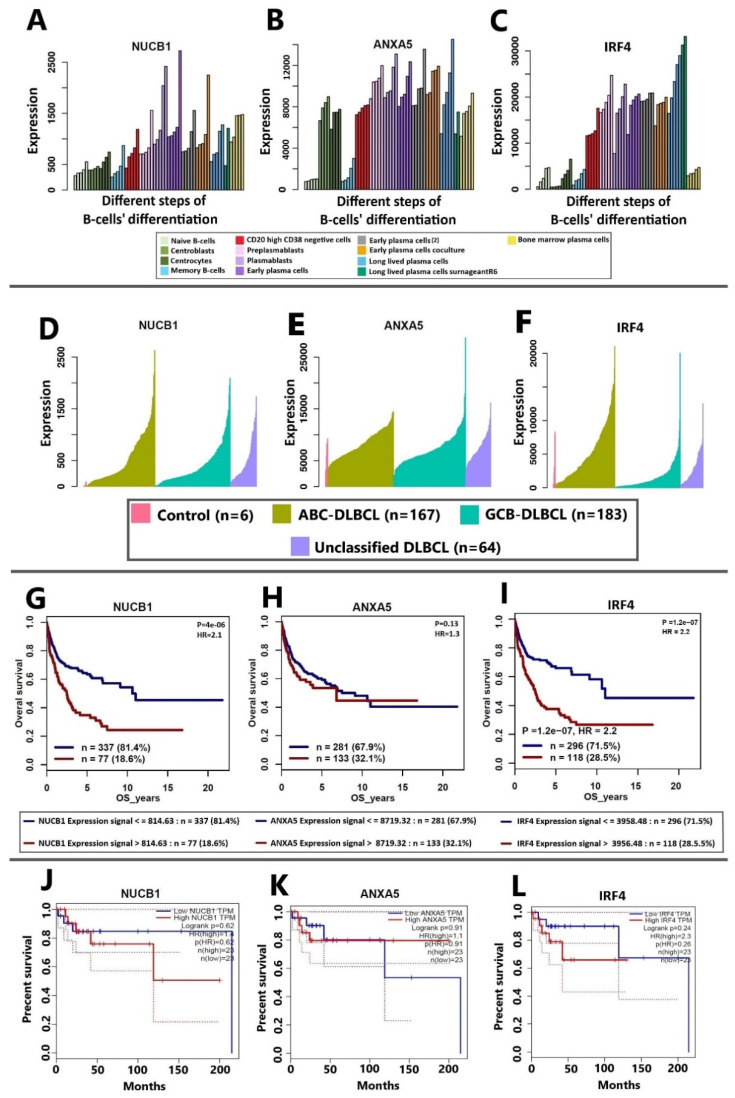
NUCB1, ANXA5, and IRF4 mRNA expression in B cell differentiation and survival analysis factored on mRNA expression. mRNA expression as fluorescence intensity summarized across probes for NUCB1 (**A**), ANXA5 (**B**), and IRF4 (**C**) in different stages of healthy B cells’ differentiation (as indicated in bottom legend). Gene expression of NUCB1 (**D**), ANXA5 (**E**), and IRF4 (**F**) in DLBCL subtypes. Overall survival plots obtained from Genomicscape factored on mRNA expression in DLBCL of NUCB1 (**G**), ANXA5 (**H**), and IRF4 (**I**). Percent survival plot generated by GEPIA based on factoring on mRNA expressions in DLBCL of NUCB1 (**J**), ANXA5 (**K**), and IRF4 (**L**).

**Figure 5 cells-12-00196-f005:**
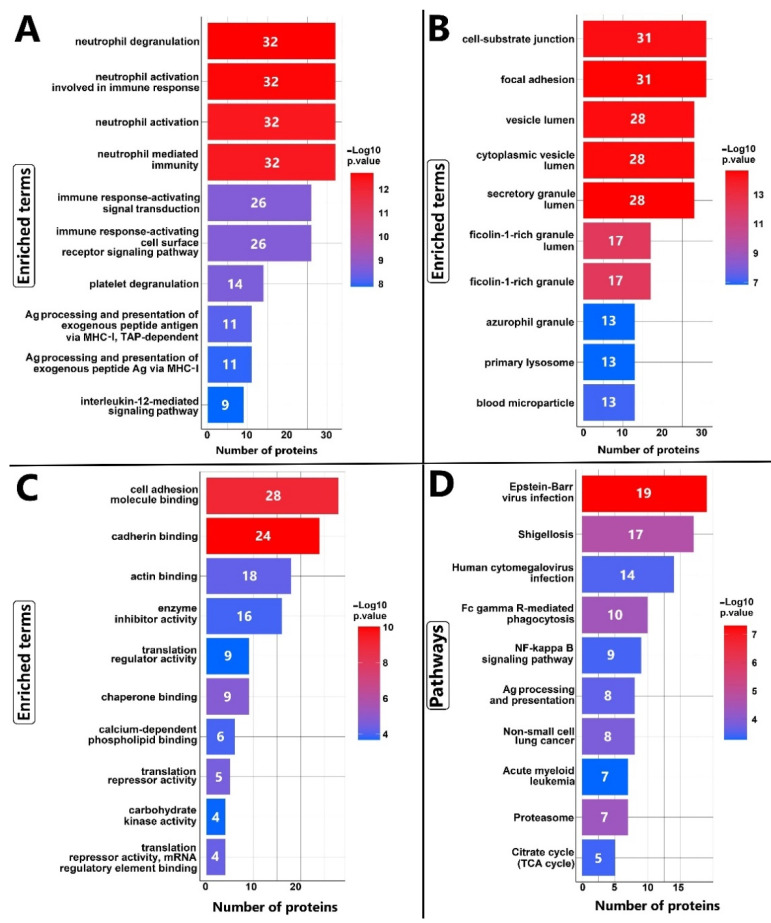
GO enrichment analysis and KEGG pathways for repeated proteins in at least four articles (n = 265). Bar plots represent the 10 most significant terms in (**A**) BP, (**B**) CC, and (**C**) MF, respectively. The bar plot (**D**) displays the top 10 most significant KEGG pathways. The number of proteins identified for each functional term is shown on the *x*-axis.

## Data Availability

The data of the current study are available on request from the corresponding author. The data are not publicly available.

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
