# Peer review of "Meta-Analysis of MS-Based Proteomics Studies Indicates Interferon Regulatory Factor 4 and Nucleobindin1 as Potential Prognostic and Drug Resistance Biomarkers in Diffuse Large B Cell Lymphoma"

_cells, 2023, doi:10.3390/cells12010196_

Round 1

Reviewer 1 Report

Review of MS. “Meta-analysis of MS-based proteomics studies indicates 3 interferon regulatory factor 4 and nucleobindin1 as potential 4 prognostic and drug resistance biomarkers in diffuse large B 5 cell lymphoma.”

This manuscript describes review of data on mass spectroscopy (MS)-based range of identified proteins [eg, interferon regulatory factor 4 (IRF4) and related dysregulated (upregulated) expression of genomic structures (annexin A5/ANXA5), nucleobindin1/NUCB1] as potential biomarkers involved in the induction of diffuse large B cell lymphoma (DLBCL) significantly affecting light zone reactions of the germinal center (LZ-GC) and cytoskeleton locomotion. 

The article is interesting in describing genomic alterations and mechanisms of action of lymph nodes growth in hematological malignancies (Hodgkin’s lymphoma-HL and non-Hodgkin’s lyphoma-NHL such as DLBCL).  The following questions and suggestions are recommended for helping authors to improve the quality of article.

A.  Major points:

1.            The authors describe MS analyses of data on gene expression profiles for three major sub-classified DLBCL, ‘activated B-like (ABC), germinal center B-cell like (GCB), and non-classified’. Antigen-induced B cells activation within germinal centers (GC) of lymphoid organs and altered expression (somatic hypermutation-SHM) of different classes of Igs and class-switch that result in dysregulation of abs profiles and in lymphomas. 

2.            There are few original series of studies on experimental models of acute and chronic inflammation that suggest direct evidence for initial antigen-inflammation-induced time-course kinetics of altered activities of immune cells (mast cells, B/plasma cells, histiocytes and GC), altered mediator expression profiles and abnormal antibody production (isozymes, IgG1/IgG2) that resulted in tissue growth, tumorigenesis and angiogenesis [1989-Massive follicular lymphoid hyperplasia in experimental chronic recurrent allergic conjunctivitis;1991-Histamine and prostacyclin: Primary and secondary release in allergic conjunctivitis; 1988-Experimental allergic conjunctivitis. Production of different isotypes of antibody by conjunctival-associated lymphoid tissue in culture]. Other related data analyses directly or indirectly support the original information [2009-Inflammation, microenvironment, and the immune system in cancer progression; 2008-Editing antigen presentation: Antigen transfer between B lymphocyte and macrophages mediated by Class A scavenger receptors; 2005-Developmental phases of inflammation-induced massive lymphoid hyperplasia and extensive changes in epithelium in an experimental model of allergy: implications for a direct link between inflammation and carcinogenesis; 2014--Chronic inflammation: Synergistic interactions of recruiting macrophages (TAMs) and eosinophils (Eos) with host mast cells (MCs) and tumorigenesis in CALTs; 2009- Inflammation, aging, and cancer: tumoricidal versus tumorigenesis of immunity: a common denominator mapping chronic diseases; 2018-Is cancer a severe delayed hypersensitivity reaction and histamine a blueprint?; 2021-Germline contribution to the immune landscape of cancer; 2022-Immune neuroplasticity (power within, adaptive, horizontal) is weakened by vaccines and drugs (power without): mitochondrial sink holes, genomic destabilization and immune disorders, etc.]

3.            It would be very helpful if authors developed a table of available data (above and related references, as cited by authors in Ms) on the role of antigen-induced unresolved inflammation that could alter tissue physiology in direction of dysregulated genetic activities/architectures, abnormal expression of mediators in local and recruited immune cells, toward tissue damage, dysregulation of B/plasma cells that potentially alter/dysregulate the genomic activities, including upregulated expression of genomic structures (annexin A5/ANXA5), nucleobindin1/NUCB1, that authors cite in immune dysregulation “… B-cell receptor signaling pathways, like NF-κB and MYD88T-cell/histiocyte-rich large B-cell lymphomas (THRLBCL), and primary central nervous system (CNS)… ABC tumor cells form during the early stages of the post‑germinal center (GC) plasma cell differentiation in the post germinal center... ABC tumor cells form during the early stages of the post‑germinal center (GC) plasma cell differentiation in the post germinal center. Previous studies suggested that ABC-DLBCLs originate from plasmablasts,… memory B cell -like phenotype as the origin ABC subtype… The transcriptional profile of the GCB sub-class transcriptionally resembles subclass is similar to that of light zone (LZ) GC B cells… the tumor cells harbour various mutations affecting chromatin-modifier genes like CREBBP and EZH2 (enhancer of zeste homolog 2)…the DLBCL tumor cells can be seeded in the testis, breast, uterus, skin, and other organs… ’’ that authors reviewed to identify immune/abs subclasses in the development of DLBCL.

B.  Minor points:

Few typos throughout the text--

Author Response

Reviewer 1:

“Review of MS. “Meta-analysis of MS-based proteomics studies indicates 3 interferon regulatory factor 4 and nucleobindin1 as potential 4 prognostic and drug resistance biomarkers in diffuse large B 5 cell lymphoma.”

This manuscript describes review of data on mass spectroscopy (MS)-based range of identified proteins [eg, interferon regulatory factor 4 (IRF4) and related dysregulated (upregulated) expression of genomic structures (annexin A5/ANXA5), nucleobindin1/NUCB1] as potential biomarkers involved in the induction of diffuse large B cell lymphoma (DLBCL) significantly affecting light zone reactions of the germinal center (LZ-GC) and cytoskeleton locomotion.

The article is interesting in describing genomic alterations and mechanisms of action of lymph nodes growth in hematological malignancies (Hodgkin’s lymphoma-HL and non-Hodgkin’s lyphoma-NHL such as DLBCL).  The following questions and suggestions are recommended for helping authors to improve the quality of article.”

AUTs: We appreciate the time and comments for improvement from reviewer 1. For clarification, we consider that IRF4, ANXA5, and NUCB1 were identified as potential diagnostic and prognostic biomarkers based on our study and not as inducers of DLBCL. We appreciate the reviewers’ suggestion to link acute and chronic inflammation to the data obtained from the MS studies. We extracted the protein markers discussed in the provided articles and compared them with the proteins identified by MS. For example, a new column was added to Table S2D showing the common immune related proteins involved in antitumor and tumorigenic function and significantly regulated proteins identified by at least 4 MS-based proteomics studies.

“A.  Major points:

  1. The authors describe MS analyses of data on gene expression profiles for three major sub-classified DLBCL, ‘activated B-like (ABC), germinal center B-cell like (GCB), and non-classified’. Antigen-induced B cells activation within germinal centers (GC) of lymphoid organs and altered expression (somatic hypermutation-SHM) of different classes of Igs and class-switch that result in dysregulation of abs profiles and in lymphomas.
  2. There are few original series of studies on experimental models of acute and chronic inflammation that suggest direct evidence for initial antigen-inflammation-induced time-course kinetics of altered activities of immune cells (mast cells, B/plasma cells, histiocytes and GC), altered mediator expression profiles and abnormal antibody production (isozymes, IgG1/IgG2) that resulted in tissue growth, tumorigenesis and angiogenesis [1989-Massive follicular lymphoid hyperplasia in experimental chronic recurrent allergic conjunctivitis;1991-Histamine and prostacyclin: Primary and secondary release in allergic conjunctivitis; 1988-Experimental allergic conjunctivitis. Production of different isotypes of antibody by conjunctival-associated lymphoid tissue in culture]. Other related data analyses directly or indirectly support the original information [2009-Inflammation, microenvironment, and the immune system in cancer progression; 2008-Editing antigen presentation: Antigen transfer between B lymphocyte and macrophages mediated by Class A scavenger receptors; 2005-Developmental phases of inflammation-induced massive lymphoid hyperplasia and extensive changes in epithelium in an experimental model of allergy: implications for a direct link between inflammation and carcinogenesis; 2014--Chronic inflammation: Synergistic interactions of recruiting macrophages (TAMs) and eosinophils (Eos) with host mast cells (MCs) and tumorigenesis in CALTs; 2009- Inflammation, aging, and cancer: tumoricidal versus tumorigenesis of immunity: a common denominator mapping chronic diseases; 2018-Is cancer a severe delayed hypersensitivity reaction and histamine a blueprint?; 2021-Germline contribution to the immune landscape of cancer; 2022-Immune neuroplasticity (power within, adaptive, horizontal) is weakened by vaccines and drugs (power without): mitochondrial sink holes, genomic destabilization and immune disorders, etc.]
  3. It would be very helpful if authors developed a table of available data (above and related references, as cited by authors in Ms) on the role of antigen-induced unresolved inflammation that could alter tissue physiology in direction of dysregulated genetic activities/architectures, abnormal expression of mediators in local and recruited immune cells, toward tissue damage, dysregulation of B/plasma cells that potentially alter/dysregulate the genomic activities, including upregulated expression of genomic structures (annexin A5/ANXA5), nucleobindin1/NUCB1, that authors cite in immune dysregulation “… B-cell receptor signaling pathways, like NF-κB and MYD88…T-cell/histiocyte-rich large B-cell lymphomas (THRLBCL), and primary central nervous system (CNS)… ABC tumor cells form during the early stages of the post‑germinal center (GC) plasma cell differentiation in the post-germinal center... ABC tumor cells form during the early stages of the post‑germinal center (GC) plasma cell differentiation in the post-germinal center. Previous studies suggested that ABC-DLBCLs originate from plasmablasts,… memory B cyell -like phenotype as the origin ABC subtype… The transcriptional profile of the GCB sub-class transcriptionally resembles subclass is similar to that of light zone (LZ) GC B cells… the tumor cells harbour various mutations affecting chromatin-modifier genes like CREBBP and EZH2 (enhancer of zeste homolog 2)…the DLBCL tumor cells can be seeded in the testis, breast, uterus, skin, and other organs… ’’ that authors reviewed to identify immune/abs subclasses in the development of DLBCL.”

AUTs: Following the reviewer’s suggestions, we now provide a discussion on based on the provided references (see discussion section). Furthermore, a column added to the table S2D outline common proteins between the immune proteins involved in antitumor and tumorigenic function and significantly regulated proteins identified by at least 4 MS-based proteomics studies.

Reviewer 2 Report

In this article, Ejtehadifar et al analysed public datasets that performed large scale proteomics/genomics/IHC based studies on DLBCL.

The article is well written, but could benefit from some improvements.

1) The authors have compared studies and  showed IRF4, ANAXA5 and NUCB1 as the main proteins reported in all studies. However, no validation is provided? Could the authors perform a western or show staining? Without validations, how are the authors making this study stand apart from all these other large scale studies?

2) Does the meta analysis identify certain type of MS analysis to be better? Can they perhaps speculate as to why the different proteomics studies show different genesets? What about correlation of co-efficient between the different MS studies? and for that between gene based and protein based studies? 

3) The way the authors describe the studies used here is confusing. Soemtimes they talk about 6 articles, other times 12 articles and so on. It would greatly benefit the reader if they would specify at the start how the studies were chosen and what subset was used for which part and why.

Author Response

In this article, Ejtehadifar et al analysed public datasets that performed large scale proteomics/genomics/IHC based studies on DLBCL.

AUTs:

We appreciate the time and comments for improvement from the reviewer. For clarification, we analyzed the literature for MS-based proteomics data sets and compared with IHC markers and genomics databases.

“The article is well written, but could benefit from some improvements.

1) The authors have compared studies and showed IRF4, ANAXA5 and NUCB1 as the main proteins reported in all studies. However, no validation is provided? Could the authors perform a western or show staining? Without validations, how are the authors making this study stand apart from all these other large scale studies?”

AUTs: Unfortunately, with the time provided for revision, it is not possible to provide experimental validations. Nevertheless, IRF4 is a widely accepted marker for poor prognosis in DLBCL, and NUCB1 was suggested as a potential marker based on six MS-based studies and survival analysis based on mRNA expression. Our aim in this study was to aggregate, in an unbiased way, current knowledge.

“2) Does the meta analysis identify certain type of MS analysis to be better? Can they perhaps speculate as to why the different proteomics studies show different genesets? What about correlation of co-efficient between the different MS studies? and for that between gene based and protein based studies?”

AUTs: We thank the reviewer for this important question. We consider that the main variations in markers identified are caused by the different types of samples analyzed (e.g., cell lines, paraffin-embedded tissue, frozen tissue, CSF, and serum). However, it is well established that MALDI and ESI ionization result in coverage of different sets of proteins. It is possible to observe that two instruments with ESI ionization but different mass analyses identify different sets of peptides. However, this does not imply that quantitative data from MS is inaccurate when samples are compared within a study. Nonetheless, the covered proteins found to be regulated vary depending on the experimental design, sample type, instrument, and setting. The comparison of proteomics and genomics revealed several similarities between MS-based proteomics and genome-based studies, although a large-scale study is still needed to compare proteomics and genomics on the same set of DLBCL samples.

“3) The way the authors describe the studies used here is confusing. Soemtimes they talk about 6 articles, other times 12 articles and so on. It would greatly benefit the reader if they would specify at the start how the studies were chosen and what subset was used for which part and why.”

AUTs:

To avoid confusion, we added a paragraph to method section 2.7. We also note that we use a lower threshold for functional enrichment analysis than for potential biomarkers. Functional enrichment analysis is robust to a few false positives and is better suited to provide an overview of many proteins.